# Toddler signaling regulates mesodermal cell migration downstream of Nodal signaling

Megan L Norris[1], Andrea Pauli[2], James A Gagnon[1], Nathan D Lord[1], Katherine W Rogers[1†], Christian Mosimann[3], Leonard I Zon[4,5,6], Alexander F Schier[1,7,8,9]*

[1]Department of Molecular and Cellular Biology, Harvard University, Cambridge, United States; [2]Research Institute of Molecular Pathology (IMP), Vienna Biocenter (VBC), Vienna, Austria; [3]Institute of Molecular Life Sciences, University of Zürich, Zürich, Switzerland; [4]Division of Hematology/Oncology, Boston Children's Hospital and Dana Farber Cancer Institute, Boston, United States; [5]Harvard Medical School, Boston, United States; [6]Stem Cell Program, Boston Children's Hospital, Boston, United States; [7]Center for Brain Science, Harvard University, Cambridge, United States; [8]The Broad Institute of Harvard and MIT, Cambridge, United States; [9]FAS Center for Systems Biology, Harvard University, Cambridge, United States

*For correspondence:
schier@fas.harvard.edu

Present address: †Systems Biology of Development Group, Friedrich Miescher Laboratory of the Max Planck Society, Tübingen, Germany

Competing interests: The authors declare that no competing interests exist.

**Abstract** Toddler/Apela/Elabela is a conserved secreted peptide that regulates mesendoderm development during zebrafish gastrulation. Two non-exclusive models have been proposed to explain Toddler function. The 'specification model' postulates that Toddler signaling enhances Nodal signaling to properly specify endoderm, whereas the 'migration model' posits that Toddler signaling regulates mesendodermal cell migration downstream of Nodal signaling. Here, we test key predictions of both models. We find that in *toddler* mutants Nodal signaling is initially normal and increasing endoderm specification does not rescue mesendodermal cell migration. Mesodermal cell migration defects in *toddler* mutants result from a decrease in animal pole-directed migration and are independent of endoderm. Conversely, endodermal cell migration defects are dependent on a Cxcr4a-regulated tether of the endoderm to mesoderm. These results suggest that Toddler signaling regulates mesodermal cell migration downstream of Nodal signaling and indirectly affects endodermal cell migration via Cxcr4a-signaling.
DOI: https://doi.org/10.7554/eLife.22626.001

## Introduction

Gastrulation is a conserved process in embryogenesis during which the three germ layers – endoderm, mesoderm and ectoderm – undergo large-scale rearrangements to shape the embryo (*Solnica-Krezel, 2005*; *Schier, 2009*; *Langdon and Mullins, 2011*; *Solnica-Krezel and Sepich, 2012*). Following their specification, endoderm and mesoderm (jointly referred to as mesendoderm) internalize beneath the ectoderm and migrate away from the site of internalization. The germ layers then converge to the dorsal side of the embryo, narrowing and extending the axis. Defects in these cellular movements can lead to a truncated body axis and abnormal organs, while defects in initial specification cause absence or reduction of mesendoderm derivatives (*Solnica-Krezel, 2005*; *Schier, 2009*; *Langdon and Mullins, 2011*; *Solnica-Krezel and Sepich, 2012*).

Germ layer specification and migration are regulated by conserved signaling pathways. Endoderm and mesoderm specification depends on the TGFβ-ligand Nodal. Secreted Nodal ligands bind to Activin receptors, which leads to Smad2 phosphorylation and target gene induction

(*Schier, 2009*). Nodal signals also trigger transcription of their own inhibitors, which belong to the Lefty family. Thus, *lefty* mutants have increased Nodal signaling resulting in increased endodermal and mesodermal gene expression and cell number during gastrulation (*Meno et al., 1999*; Rogers et al., unpublished ).

After specification by Nodal, endodermal cells are tethered to mesodermal cells through a fibronectin-integrin link genetically regulated by the GPCR Cxcr4a (*Mizoguchi et al., 2008*; *Nair and Schilling, 2008*). Loss of Cxcr4a releases this tether and results in excessive animal pole-directed migration of endodermal cells, while mesodermal cell migration is unaffected (*Mizoguchi et al., 2008*; *Nair and Schilling, 2008*).

An additional GPCR pathway, regulated by Toddler/Apela/Elabela, is also required for proper endoderm and mesoderm formation and migration (*Chng et al., 2013*; *Pauli et al., 2014*). In this pathway, the secreted peptide Toddler, which is highly conserved throughout vertebrates, signals via the GPCR APJ (in zebrafish: Apelin receptor A and B, jointly referred to as Apelin receptor). In *toddler* mutants, initial specification of endoderm and mesoderm is normal (*Pauli et al., 2014*), but by mid-gastrulation, *toddler* mutants have fewer endodermal cells and mesendodermal cell migration is reduced (*Chng et al., 2013*; *Pauli et al., 2014*). *toddler* mutants generally die around 7 days post fertilization (dpf) with deformed hearts, blood accumulation, edema and endodermal abnormalities (*Chng et al., 2013*; *Pauli et al., 2014*).

Two non-exclusive models have been proposed for the role of Toddler signaling in gastrulation. One model, the 'specification model', postulates that Toddler's primary role is to promote the specification of endoderm, which when defective leads to abnormal migration of mesendodermal cells (*Chng et al., 2013*). This model is supported by the observation of fewer endodermal cells in *toddler* mutants. Moreover, Apelin receptor signaling has been proposed to enhance Nodal signaling, possibly accounting for the endoderm cell number defects in *toddler* mutants (*Deshwar et al., 2016*). Based on these findings, it has been proposed that Toddler signaling enhances Nodal signaling, allowing for proper endoderm specification, which in turn promotes mesendodermal cell migration.

An alternative model for Toddler signaling, the 'migration model', postulates that the primary role of Toddler signaling is to regulate mesendodermal cell migration. This model is supported by the observation that mesendodermal cells migrate more slowly during internalization in *toddler* mutants, and that *apelin receptor a* and *b* gene expression depend on Nodal signaling (*Tucker et al., 2007*; *Pauli et al., 2014*). These findings place Toddler signaling downstream of Nodal signaling and endoderm specification and suggest a primary role for Toddler signaling in mesendodermal cell migration.

To clarify how Toddler regulates gastrulation, we tested four aspects of the specification and migration models. First, we determined if the defects in *toddler* mutants result primarily from reduced endoderm specification. Second, we analyzed if *toddler* mutants display reduced Nodal signaling. Third, we examined how migration of mesodermal cells is affected in *toddler* mutants. Fourth, we tested if Toddler's primary site of action is endoderm, mesoderm, or both cell types. We found that reduced endoderm specification is not sufficient to explain the *toddler* mutant phenotype, that Nodal signaling initiates normally in *toddler* mutants, and that Toddler signaling acts on mesodermal cells to allow animal pole-directed migration. Our results support a modified version of the migration model in which Toddler signaling acts downstream of Nodal signaling to regulate mesodermal cell migration while indirectly regulating endodermal cell migration via Cxcr4a signaling.

## Results

### Increased endoderm specification does not rescue *toddler* mutants

In *toddler* mutants, endoderm initially appears normal (*Pauli et al., 2014*), but by mid-gastrulation, the number of endodermal cells is reduced (*Figure 1A–B*)(*Chng et al., 2013*; *Pauli et al., 2014*). Since Toddler has been implicated in endodermal specification (*Chng et al., 2013*; *Ho et al., 2015*), we revisited the cause of reduced endodermal cell numbers in *toddler* mutants. We found that endodermal cell numbers are comparable between wild-type and *toddler* mutant embryos at 60% epiboly and that they go on to divide at similar rates, suggesting that initial specification and subsequent proliferation are not affected (*Figure 1—figure supplement 1*)(*Pauli et al., 2014*). Instead,

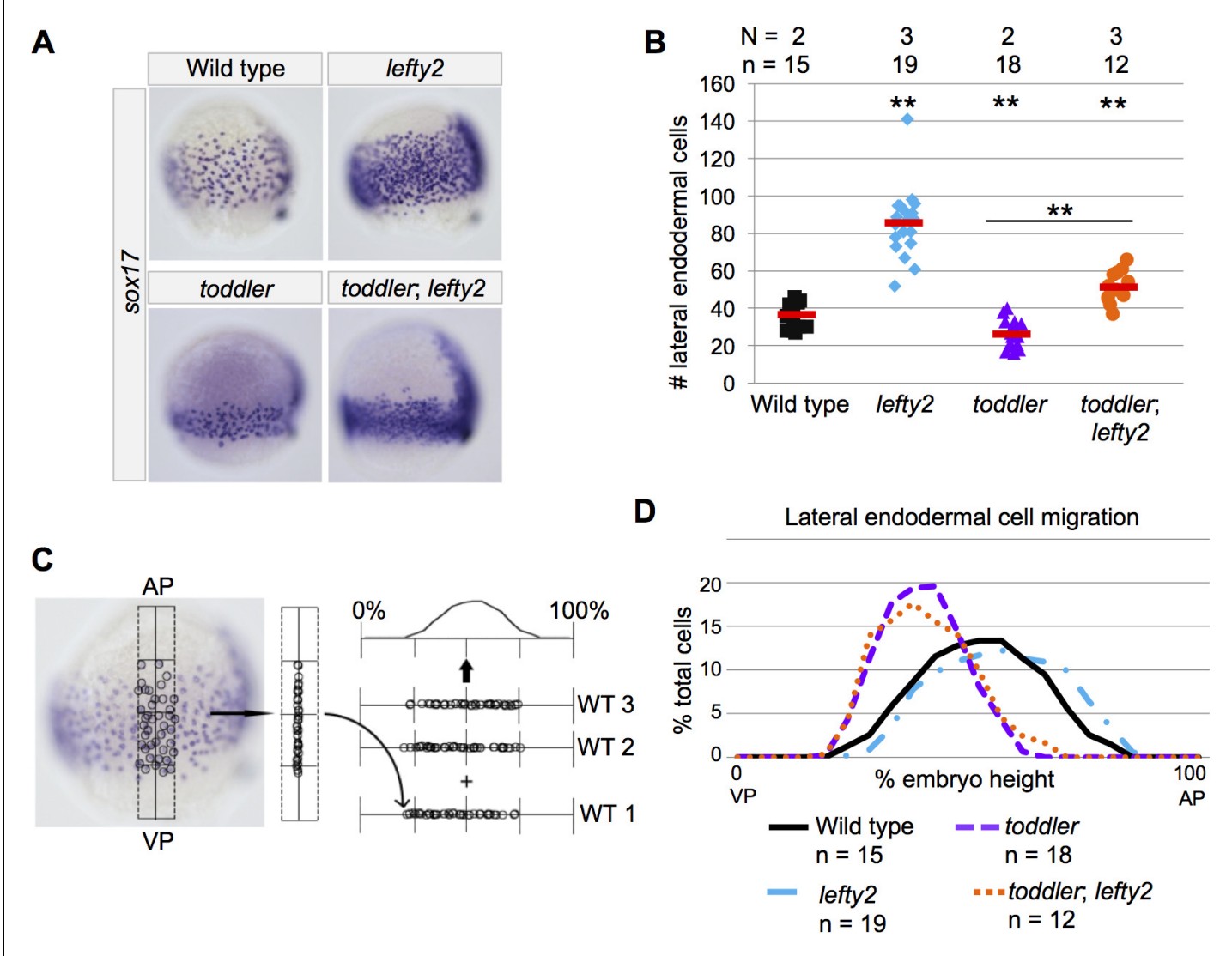

**Figure 1.** Increased endodermal specification does not rescue endodermal cell migration in *toddler;lefty2* double mutants. (A) Representative images of embryos analyzed in B and D. In situ hybridization with a *sox17* probe at 75% epiboly; dorsal to the right. Endoderm in *toddler;lefty2* double mutants resembles *toddler* single mutants. (B) The number of lateral endodermal cells is increased in *toddler;lefty2* double mutants compared to wild-type and *toddler* single mutant embryos. Each point represents a single embryo. Red bars are averages. p-Values for pairwise comparison with wild type unless otherwise noted. **p<0.001; unpaired two-tailed t-test. N = number of independent experiments; n = number of embryos. (C) Schematic representation of experimental measurements shown in B and D. The locations of lateral endodermal cells in individual embryos are measured relative to the AP-VP axis and then consolidated across embryos. (D) Measurement of frequency with which cells were found at a given location in an embryo of a certain genotype. A cell at the animal pole corresponds to 100% embryo height, while a cell at the vegetal pole corresponds to 0%. AP = Animal pole; VP = vegetal pole. The same embryos were measured in B and D.

DOI: https://doi.org/10.7554/eLife.22626.002

The following figure supplement is available for figure 1:

**Figure supplement 1.** Initial specification of endoderm is normal in *toddler* mutants.

DOI: https://doi.org/10.7554/eLife.22626.003

we found increased rates of cell death in *toddler* mutant embryos, including endodermal cell death, which may account for the decreased cell numbers observed during later gastrulation (*Figure 1—figure supplement 1*).

If the deficit in endodermal cells is the primary defect in *toddler* mutants, increasing endodermal cell number might rescue other aspects of the *toddler* mutant phenotype. To increase endodermal cell number, we used *lefty2* mutants, which have more endodermal cells due to increased Nodal

signaling (*Figure 1A–B*)(*Meno et al., 1999*; *Agathon et al., 2001*; Rogers et al., ). We analyzed endoderm specification and mesendodermal cell dispersion in *toddler;lefty2* double mutants and compared it to *toddler* and *lefty2* single mutants and wild-type embryos (*Figures 1* and *2*). As expected, *lefty2* single and *toddler;lefty2* double mutant embryos had more endodermal cells than *toddler* single mutants and wild-type embryos (*Figure 1A–B*). While endodermal cells in *lefty2* single mutants migrated normally, endodermal cell dispersion in *toddler;lefty2* double mutants was not rescued and resembled *toddler* single mutant siblings (*Figure 1A,C and D*). As with endoderm, mesodermal cell dispersion was not rescued in *toddler;lefty2* double mutants (*Figure 2A–B*). Thus, increasing endodermal cell number did not rescue the endodermal or mesodermal cell migration defects in *toddler* mutants.

To test if increased endodermal cell number can rescue later phenotypes found in *toddler* mutants, we analyzed embryo morphology and heart size in 2- to 3-day-old *toddler;lefty2* double mutants. Hearts in *toddler;lefty2* double mutants resembled *toddler* single mutants, except in rare cases when hearts were wild-type in size (*Figure 2C–D*). By 3 dpf, *toddler;lefty2* double mutant larvae had similar or more severe patterning defects as compared to their *toddler* single mutant siblings (*Figure 2E–F*). Taken together, these results indicate that increasing endoderm cell number cannot rescue *toddler* mutant phenotypes.

## Nodal signaling is established normally in *toddler* mutants

The specification model suggests that Toddler/Apelin receptor signaling enhances Nodal signaling (*Chng et al., 2013*; *Deshwar et al., 2016*). To test this prediction, we used four assays to analyze Nodal signaling in *toddler* mutants:

First, we used in situ hybridization to examine expression of two direct Nodal target genes, *lefty1* and *lefty2*, just before gastrulation and found no difference between wild-type and *toddler* mutant embryos (*Figure 3A*). Similarly, *apelin receptor a* and *b* single and double mutants also displayed normal expression of the Nodal target genes *lefty1* and *lefty2* (*Figure 3B–C*).

Second, we analyzed Nodal target gene expression by RNA-Seq at four time points prior to or during early gastrulation in wild-type and *toddler* mutants. We found no significant differences at any time-point in the expression of Nodal target genes (*Dubrulle et al., 2015*) or broadly across all genes (*Figure 3D*)(*Figure 3—Source data 1*).

Third, wild-type and *toddler* mutant embryos were injected at the one cell stage with increasing levels of Nodal mRNA and Nodal target gene expression was measured just prior to gastrulation using qRT-PCR. We found no significant difference in the ability of wild-type and *toddler* mutants to respond to exogenous Nodal (*Figure 4A*).

Fourth, cells expressing Nodal mRNA were transplanted into a host embryo and the host was assayed for Nodal target gene induction using in situ hybridization (*Figure 4B*). Wild-type and *toddler* mutants displayed target gene expression at the same frequencies and across the same distances (*Figure 4C–D*). These results indicate that the establishment of Nodal signaling is normal in *toddler* mutants.

## Toddler receptors are expressed in mesodermal cells

Toddler signaling has primarily been investigated in the context of endoderm development (*Chng et al., 2013*; *Pauli et al., 2014*), but mesoderm also displays migration defects in *toddler* mutants (*Pauli et al., 2014*). Furthermore, *apelin receptor a* and *b* gene expression in wild-type embryos resembles expression of the mesoderm markers *fn1a* and *fascin,* respectively, and is dependent on signaling by the mesendoderm inducer Nodal (*Pauli et al., 2014*). To further analyze *apelin receptor a* and *b* gene expression in endoderm versus mesoderm, we created embryos with excess or diminished levels of endoderm by manipulating the endoderm master regulator *sox32* with mRNA injection or CRISPR/Cas9-mediated mutagenesis (*Figure 5A–B*, *Figure 5—figure supplement 5* ) (*Kikuchi et al., 2001*). As expected, the endodermal marker *sox17* displayed increased expression in *sox32* mRNA-injected embryos or decreased expression in *sox32* mutant embryos (*Figure 5A*). In contrast, the expression of mesodermal markers *fn1a* and *ta* or of *apelin receptor a* and *b* increased upon loss of endoderm (*Figure 5A* and *Figure 5—figure supplement 5* ). Furthermore, *apelin receptor b* expression was retained at the site of internalization in embryos lacking

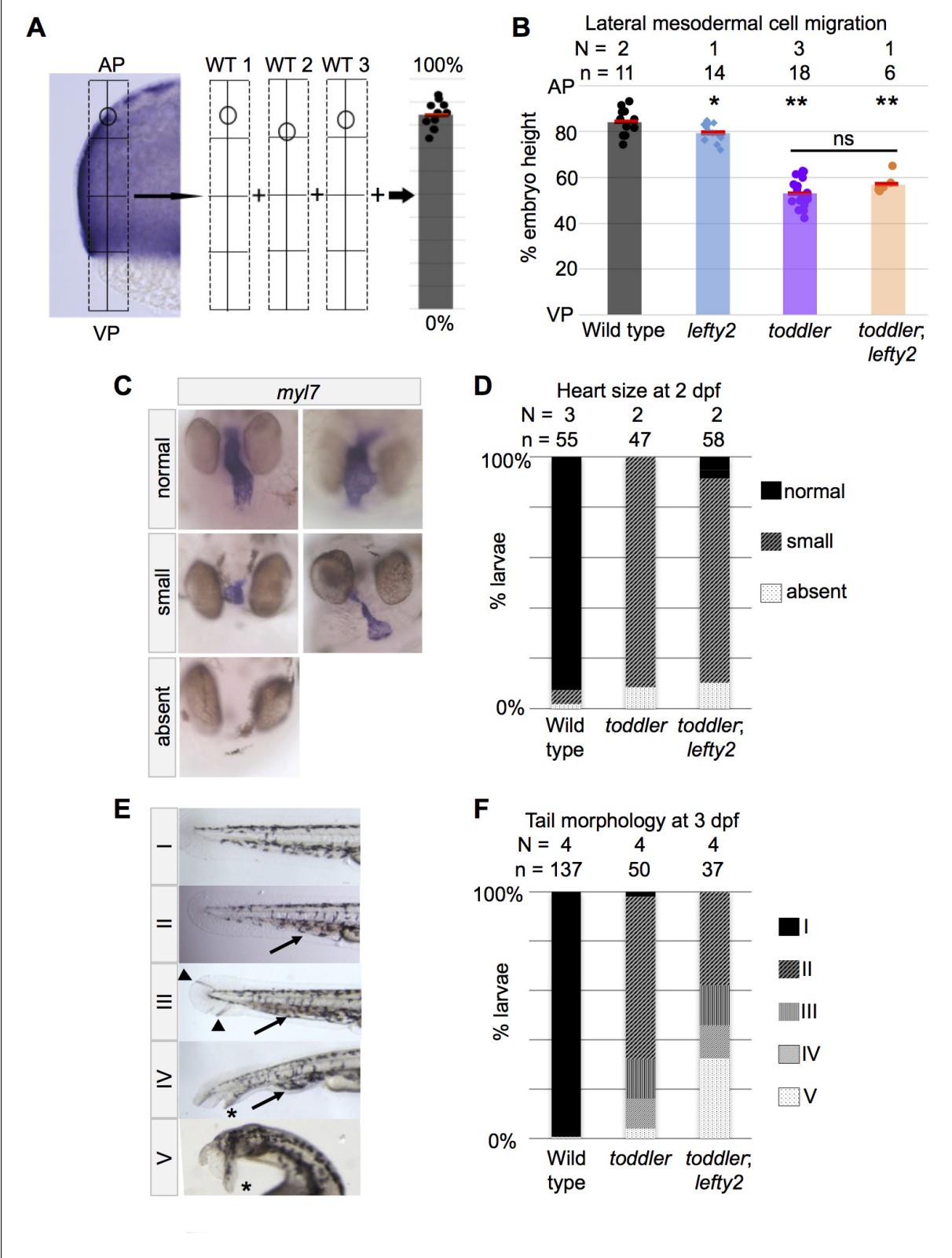

**Figure 2.** Mesodermal cell migration and larval phenotypes are not rescued by increased numbers of endodermal cells. (A) Schematic representation of experimental measurements shown in B. Densely packed, *fn1a* + lateral mesodermal cells are viewed by cross section. Location of the most animally migrated lateral cells of individual embryos are measured relative to AP-VP axis, then consolidated across embryos. AP = Animal pole; VP = vegetal pole. (B) Defects in animal-pole directed migration of mesodermal cells are still present in *toddler;lefty2* double mutants. Each point

*Figure 2 continued on next page*

*Figure 2 continued*

represents a single embryo. Red bars are averages. p-Values for pairwise comparison with wild type unless indicated otherwise. *p=0.04, **p<0.001; ns: p=0.08; unpaired two-tailed t-test. (**C–D**) *toddler;lefty2* double mutants resemble *toddler* single mutant siblings in respect to heart phenotypes at 2 days post fertilization (dpf). Hearts were classified as small if shortened by more than half of normal length and/or excessively narrow or thin. Hearts that were neither thin nor short but appeared to have looping defects or other patterning defects were classified as normal. Heart phenotypes were scored blind to genotype for *toddler* and *toddler;lefty2* mutants. (**C**) Representative images of phenotypic classes after in situ hybridization for *myl7*. (**D**) Quantification of C. (**E–F**) *toddler;lefty2* double mutants are more poorly patterned than *toddler* single mutant siblings in respect to tail phenotypes at 3 dpf. (**E**) Representative images of phenotypic classes. Phenotype classes II-V lack circulation. Arrow: accumulated blood. Arrowhead: defects in mesenchyme. * Duplicated tail tip. (**F**) Quantification of E. (**B,D,F**) N = number of independent experiments; n = number of embryos.

DOI: https://doi.org/10.7554/eLife.22626.004

endoderm, suggesting its expression is not ectodermal but mesodermal (*Figure 5B*). These results indicate that *apelin receptor a* and *b* are expressed in mesodermal progenitors.

## Toddler regulates animal-pole directed movement of mesodermal cells independent of endoderm

To determine what aspects of mesodermal cell migration are affected in *toddler* mutants, we tracked migrating *drl:eGFP*-positive ventrolateral mesodermal cells (*Mosimann et al., 2015*) in wild-type and *toddler* mutant embryos during gastrulation using light sheet microscopy (*Figure 5C* and *Video 1*). We found that mesodermal cells moved more slowly in *toddler* mutants than in wild-type embryos (*Figure 5D*). In addition, *toddler* mutant mesodermal cells showed diminished animal pole-directed migration, resulting in persistent vegetal and dorsal-ward migration during gastrulation (*Figure 5E*). Taken together, these results indicate that Toddler signaling is required for proper animal-pole directed movement of mesodermal cells during gastrulation.

To test the role of endodermal cells in Toddler's regulation of mesodermal cell migration, we blocked endoderm specification by mutating *sox32* and then analyzed mesoderm dispersion. We found that lack of endoderm did not change the normal mesoderm dispersion in wild-type embryos or the reduced migration in *toddler* mutants (*Figure 6A–B*). These results indicate that the mesodermal cell migration defects in *toddler* mutants are independent of endoderm and suggest a direct role of Toddler signaling on mesoderm.

## Endodermal cell migration does not require Toddler signaling in the absence of Cxcr4a

During gastrulation, endoderm is physically tethered to mesoderm (*Mizoguchi et al., 2008*; *Nair and Schilling, 2008*). Thus, Toddler might act directly upon mesoderm and only indirectly affect endoderm via the tether. To test this hypothesis, we severed the tether linking endodermal cells to mesodermal cells by removing Cxcr4a. We used Cas9-mediated mutagenesis to create a null mutant for *cxcr4a* (as described in Materials and methods) and observed that mesodermal cells migrated normally (*Figure 7A–B*), while the most vegetal endodermal cells dispersed further animally than in wild type, consistent with previous Cxcr4a morphant studies (*Figure 7C–D*) (*Mizoguchi et al., 2008*; *Nair and Schilling, 2008*). As expected, mesodermal cell migration in *toddler;cxcr4a* double mutants was as reduced as in *toddler* single mutants (*Figure 7A–B*). Strikingly, however, endoderm dispersion was increased in *toddler;cxcr4a* double mutants as compared to *toddler* single mutants (*Figure 7C–D*). While endoderm migration was increased in *toddler;cxcr4a* double mutants, the number of endodermal cells did not change (*Figure 7E*). These observations indicate that the reduced mesodermal cell migration in *toddler* mutants limits endodermal dispersion and that Toddler signaling is not required for endodermal cell migration per se.

## Discussion

Here, we clarify the role of Toddler signaling during gastrulation by reporting five main observations. First, increased endoderm specification in *toddler;lefty2* double mutants does not alleviate *toddler* mutant defects (*Figures 1* and *2*). Second, *toddler* mutants initiate endogenous Nodal target gene expression normally and respond indistinguishably from wild type to exogenous Nodal sources (*Figures 3* and *4*). Third, Toddler signaling is required for proper animal pole-directed migration of

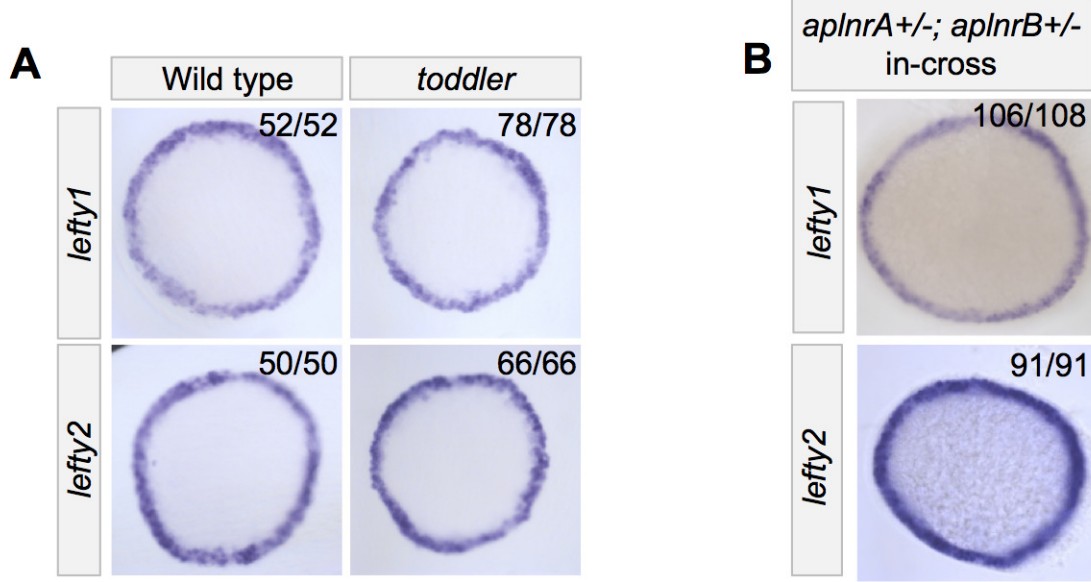

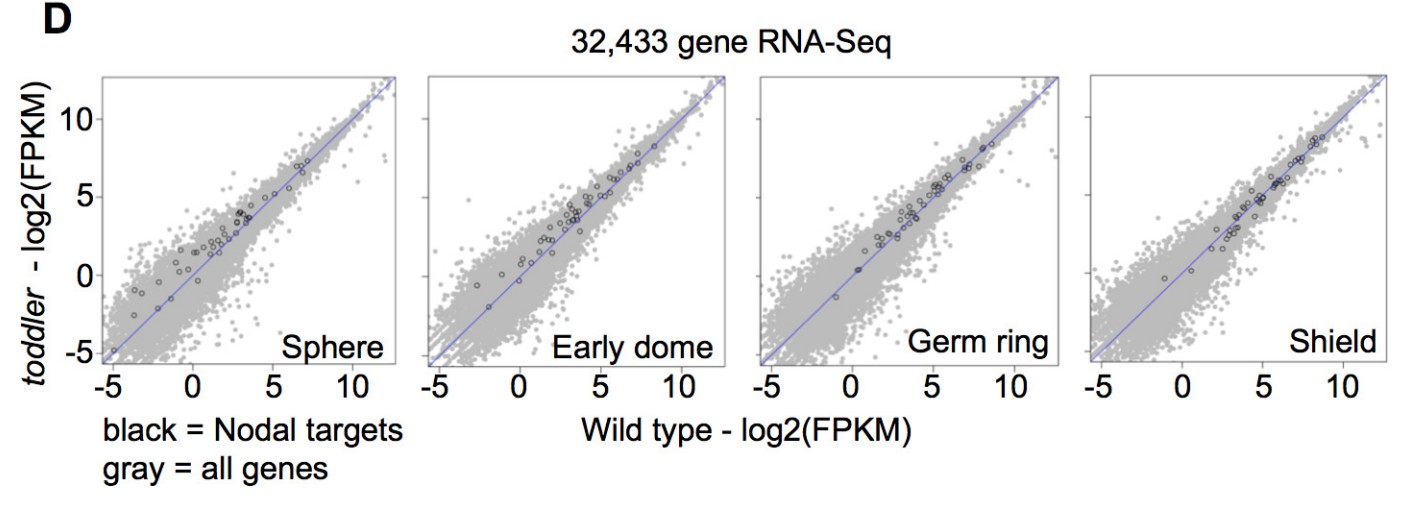

**C**

Embryos with staining matching that in (B) at 50% epiboly from *aplnrA +/-; aplnrB +/-* parents

| | Wild type | *aplnrA -/-* | *aplnrB -/-* | *aplnrA -/-; aplnrB +/-* | *aplnrA +/-; aplnrB -/-* | *aplnrA -/-; aplnrB -/-* |
|---|---|---|---|---|---|---|
| *lefty1* | 11/11 | 4/5 | 5/5 | 13/13 | 10/10 | 6/6 |
| *lefty2* | 5/5 | 6/6 | 10/10 | 6/6 | 12/12 | 9/9 |

**Figure 3.** Wild-type and *toddler* mutant embryos have similar levels of Nodal signaling. (A) Representative images of Nodal target gene expression in wild-type and *toddler* mutant embryos based on in situ hybridization at 50% epiboly. Animal pole facing up. Three biological replicates. (B–C) *aplnrA;* *aplnrB* double heterozygote parents were incrossed and their progeny examined by in situ hybridization at 50% epiboly. (B) Representative images of Nodal target gene expression in offspring from *aplnrA;aplnrB* double heterozygote parents. Animal pole facing up. (C) All embryos from B were genotyped. Results for relevant genotypes are shown. Expression of *lefty1* and *lefty2* in *aplnrA;aplnrB* double mutant embryos was indistinguishable
*Figure 3 continued on next page*

*Figure 3 continued*

from wild type. (D) RNA-sequencing before and during early gastrulation reveals no differential transcription of Nodal targets (black dots) between wild-type and *toddler* mutant embryos. All other genes are shaded in light gray. See also *Figure 3—Source data 1*.

DOI: https://doi.org/10.7554/eLife.22626.005

The following source data is available for figure 3:

**Source data 1.** Gene information for variable genes in RNA-Seq data set.

DOI: https://doi.org/10.7554/eLife.22626.006

mesoderm during gastrulation (*Figure 5*). Fourth, the defects in mesodermal cell migration are independent of endoderm in *toddler;sox32* double mutants (*Figure 6*). Fifth, untethering endodermal and mesodermal cells in *toddler;cxcr4a* double mutants improves endodermal cell migration without affecting mesoderm (*Figure 7*).

These results and previous studies support the following model for Toddler function. Nodal signaling induces the expression of *apelin receptor a* and *b* in mesendodermal cells, rendering them sensitive to Toddler peptide (*Figure 5A–B*)(*Pauli et al., 2014*; *Tucker et al., 2007*). Toddler signaling directly regulates animal-pole directed migration of mesodermal cells (*Figures 5* and *6*) (*Pauli et al., 2014*). A Cxcr4a-regulated tether between endodermal and mesodermal cells allows Toddler signaling to indirectly affect endodermal cell migration (*Figure 7*)(*Mizoguchi et al., 2008*; *Nair and Schilling, 2008*).

Our findings about Toddler signaling differ from a recent study that concluded Apelin receptor signaling enhances Nodal signaling (*Deshwar et al., 2016*). Whereas Deshwar and colleagues reported that *apelin receptor a and b* double morphants appear to have a delayed onset of Nodal signaling, we find that *toddler* mutants as well as *apelin receptor a* and *b* single and double mutants establish Nodal signaling normally (*Figure 3*). There are multiple possibilities that may explain these different results. First, it is possible that Toddler and/or the Apelin receptors have independent functions (*Ho et al., 2015*). Second, although apelin receptor mutants and morphants have similar morphological defects (*Deshwar et al., 2016*), knockdown of the receptors via morpholinos might reveal a phenotype that is masked or compensated in the null mutants (*Rossi et al., 2015*; *Wei et al., 2017*). Third, morpholino injection may cause non-specific delayed development or toxicity, as suggested by the epiboly defects seen in *apelin receptor a* morphants but not *apelin receptor a* mutants (*Ekker and Larson, 2001*; *Kok et al., 2015*; *Deshwar et al., 2016*).

Our study indicates that Toddler is a regulator of mesodermal cell migration and only indirectly regulates endodermal cell migration during gastrulation. Accordingly, we find both similarities and differences in the role of Toddler signaling in the migration of endoderm versus mesoderm. Both endodermal cells (*Pauli et al., 2014*) and mesodermal cells (this study, *Figure 5D*) migrate more slowly in the absence of Toddler signaling. However, we find that *toddler* mutant mesodermal cells have reduced animal pole-directed migration during gastrulation. This is in contrast to endodermal cells, which initially undergo a random walk, including animal pole-directed movement, in wild-type and *toddler* mutants (*Pauli et al., 2014*). The intrinsic ability of endoderm to migrate without Toddler signaling is fully revealed in the abundant animal pole-directed migration of endoderm in *toddler;cxcr4a* double mutants. Thus, the ability to undergo animal pole-directed migration is retained in endodermal cells but compromised in mesodermal cells in the absence of Toddler signaling.

These findings raise important questions for future studies. Our work and others have suggested an anti-apoptotic function for Toddler (*Ho et al., 2015*), raising the question of whether the role for Toddler in cell survival is related to or independent of its role in cell migration? How does Toddler signaling regulate mesodermal cell migration? Our RNA-Seq results in *toddler* mutants did not reveal major changes in gene expression during gastrulation (*Figure 3*), suggesting that Toddler's primary function may be post-transcriptional. The identities of such potential effectors remain unknown. Equally unknown is the nature of the regulators of endodermal cell migration. In the absence of both Toddler and Cxcr4a signaling, endoderm migration is surprisingly normal. What regulates endoderm migration in this background? Toddler signaling will continue to provide an entry point to address these questions.

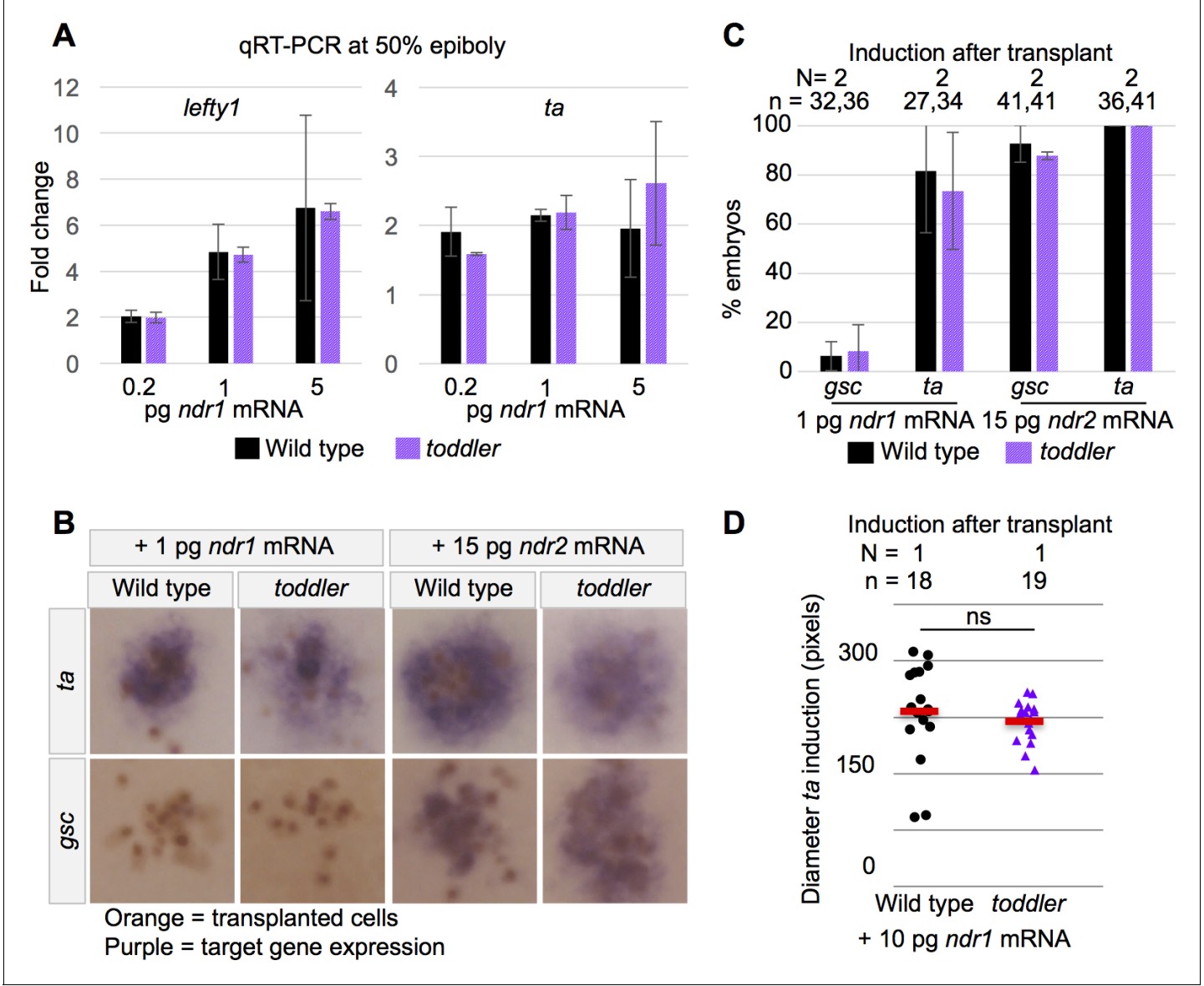

**Figure 4.** Wild-type and *toddler* mutant embryos respond similarly to exogenous Nodal sources. (A) qRT-PCR at 50% epiboly on embryos injected at the one-cell stage with water or increasing levels of Nodal mRNA. Fold change in expression is relative to water control. Two biological replicates. (B–D) Clones of cells expressing exogenous Nodal mRNA and GFP were transplanted into host embryos at sphere stage and collected for in situ hybridization 1.5 hr later. Donors and hosts were always of matching genotype. (B) Representative images of quantifications in C. Orange staining marks anti-GFP labeled transplanted cells. Purple staining is from in situ hybridization for a Nodal target gene. (C) Percentage of embryos for which Nodal target gene expression was visible via in situ hybridization. N = number of independent experiments. n = number of embryos. (D) Diameter of induction of *ta* expression around the clone of transplanted cells. Each point represents a single embryo. Red bars are averages; ns: p=0.44; unpaired two-tailed t-test. (A and C) Means ±SEM.

DOI: https://doi.org/10.7554/eLife.22626.007

# Materials and methods

## Zebrafish lines and husbandry

Zebrafish, including TLAB wild-type fish, were maintained according to standard protocols. *toddler* double mutant lines were obtained by crossing *toddler* mutants to either *lefty1*[a154], *lefty2*[a146] or *cxcr4a* mutants. *toddler* homozygous/*lefty1*[a154] heterozygous, *toddler* homozygous/*lefty2*[a146] heterozygous and *toddler* homozygous/*cxcr4a* heterozygous fish were obtained by injecting 2 pg

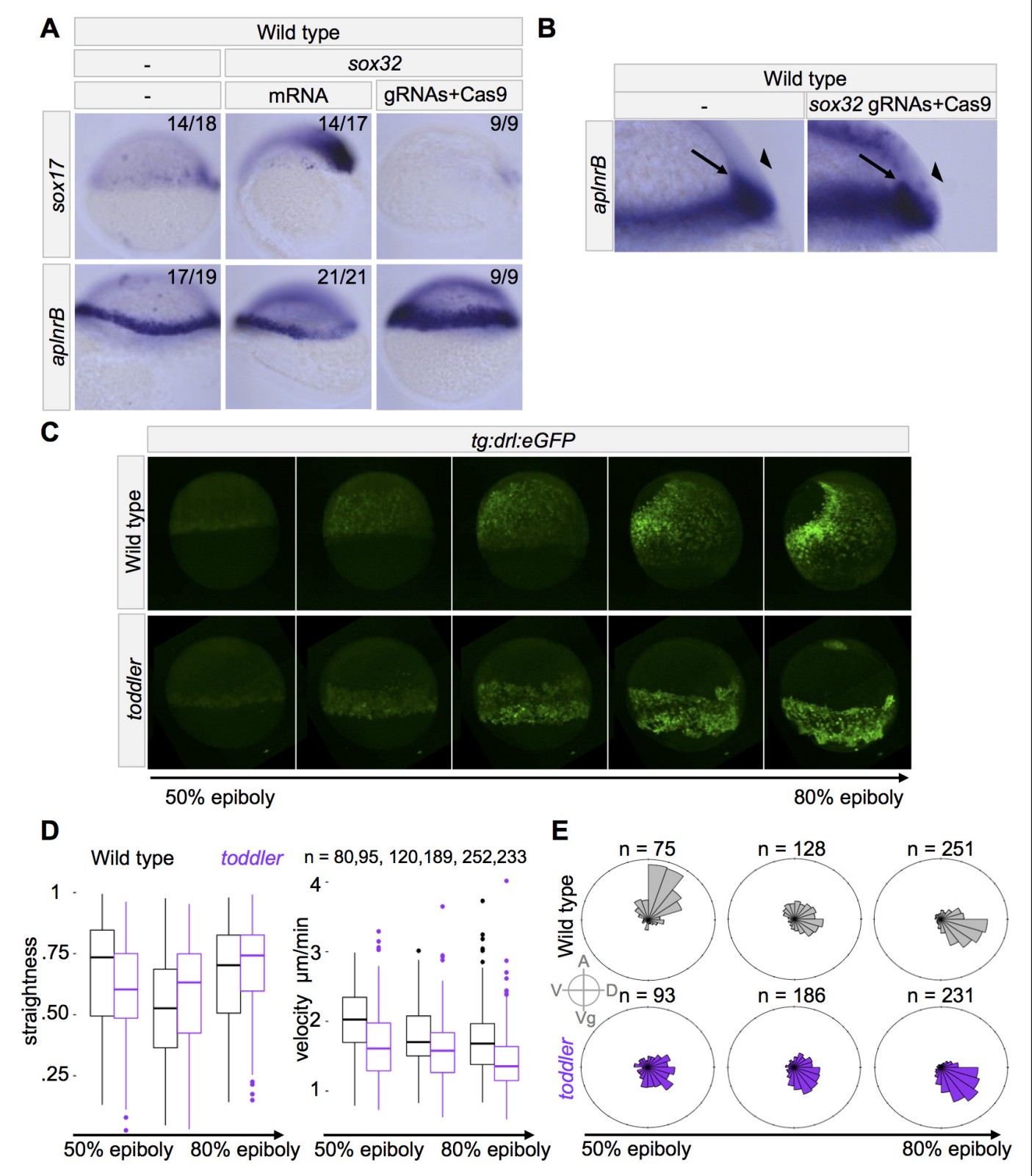

**Figure 5.** *toddler* mutant mesodermal cells have diminished animal pole-directed migration during gastrulation. (A–B) Cas9-mediated mutagenesis was used to generate *sox32* mutants. Embryos were injected at the one-cell stage either with *sox32* mRNA or with Cas9 + *sox32* gRNAs. in situ hybridization for *sox17* and *aplnrB* at shield stage; dorsal to the right. (A) Expression of *aplnrB* decreases in the presence of excess endoderm and increases in the absence of endoderm. (B) Cross-section of shield stage embryo. The majority of cells expressing *aplnrB* in embryos lacking endoderm
*Figure 5 continued on next page*

*Figure 5 continued*

are internalized and unlikely to be ectodermal. Arrow: internalized cells. Arrowhead: ectoderm. (C–E) Mesodermal cell migration was tracked during gastrulation using a *drl:eGFP* transgene and lightsheet microscopy. *drl:eGFP* labels ventrolateral mesoderm during gastrulation. *toddler* mutant measurements represent three embryos. Wild-type measurements represent two wild-type and one heterozygous embryos. (C) Representative still frames of maximum intensity projections from a wild-type and *toddler* mutant *drl:eGFP* transgenic embryo. See also *Video 1*. (D–E) Movies spanning 50% to 85% epiboly were aligned at the onset of internalization and binned into 45 min windows. (D) Measurement of straightness and velocity of mesodermal cells in wild-type and *toddler* mutants. Straightness is defined as the difference between the total movement of a cell in all directions divided by the net displacement (the actual distance between the cell's beginning and ending location). (E) *toddler* mutant cells have diminished migration animally during gastrulation. Each bin represents the proportion of cells moving in a given direction weighted by the total distance traveled in that direction. n = number of cells. A = animal, D = dorsal, Vg = vegetal, V = ventral.

DOI: https://doi.org/10.7554/eLife.22626.008

The following figure supplement is available for figure 5:

**Figure supplement 1.** Apelin receptor A is expressed in mesodermal cells.
DOI: https://doi.org/10.7554/eLife.22626.009

*toddler* mRNA at the one cell stage as previously described (*Pauli et al., 2014*). Embryos were raised at 28°C and staged by morphology unless otherwise noted. *lefty1*[a145] is a null allele harboring a 13 base pair deletion that destroys a PshAI restriction enzyme site (Rogers et al. submitted). *lefty2*[a146] is a null allele harboring an 11 base pair deletion that was genotyped with allele-specific primers (*Table 1*)(Rogers et al. submitted). *aplnrB* mutants harbor a 1 bp deletion in the 52nd amino acid, which creates an AflIII restriction enzyme cleavage site. A stable *cxcr4a* mutant line was generated using Cas9-mediated mutagenesis described below. The *cxcr4a* mutant line harbors a 433 bp deletion of a 1083 bp long coding sequence, generating a frameshift after the 28th amino acid and was genotyped with allele-specific primers (see *Table 1* for gRNAs and primers). *toddler* mutants and *aplnra*[mu296] mutants were genotyped as previously described (*Pauli et al., 2014*; *Helker et al., 2015*). See *Table 1* for relevant primer sequences.

## RNA and gRNA synthesis, injection and genotyping

Capped mRNA for *sox32* was transcribed from linearized plasmid using the T3 mMessage machine kit from Ambion (Pittsburgh, Pennsylvania). mRNA was injected at the one cell stage.

Five gRNAs targeting *cxcr4a* and nine gRNAs targeting *sox32* were designed using ChopChop (*Montague et al., 2014*) and transcribed as described in *Gagnon et al. (2014)*. The five *cxcr4a* gRNAs were pooled and injected with Cas9 protein at the one cell stage. Injected embryos were raised to adulthood and monitored for germline transmission by genotyping individual F1 embryos with primers spanning the targeted region of the gene. Nine *sox32* gRNAs were pooled and injected with Cas9 protein at the one-cell stage. A subset of embryos were collected for in situ hybridization at shield and 75% epiboly while others were raised to 3 dpf to confirm the expected gross morphological phenotype. See *Table 1* for genomic target sequences for gRNAs.

Whole mount in situ hybridization was carried out as previously described (*Thisse and Thisse, 2008*). Stained embryos were imaged in glycerol or were dehydrated in methanol and cleared in BB/BA. Embryos were imaged on an Axio Imager.Z1 microscope. For genotyping after imaging, embryos were washed in water before isolating genomic DNA.

## Quantification of cell location and cell number

Lateral endodermal and mesodermal cells were analyzed in embryos at 70–85% epiboly, except for *Figure 1* Supplement, when they were

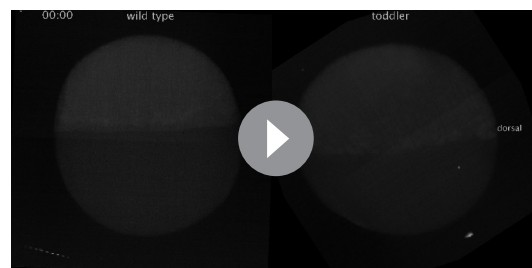

**Video 1.** Toddler regulates animal-poled directed migration of mesodermal cells. Embryo pair shown in *Figure 5C*. Embryos are transgenic for *drl:eGFP,* which labels ventrolateral mesoderm during gastrulation. Embryos are aligned at the onset of internalization. Maximum intensity projections from a light sheet microscope spanning 50% to 85% epiboly, time in hr: min.
DOI: https://doi.org/10.7554/eLife.22626.010

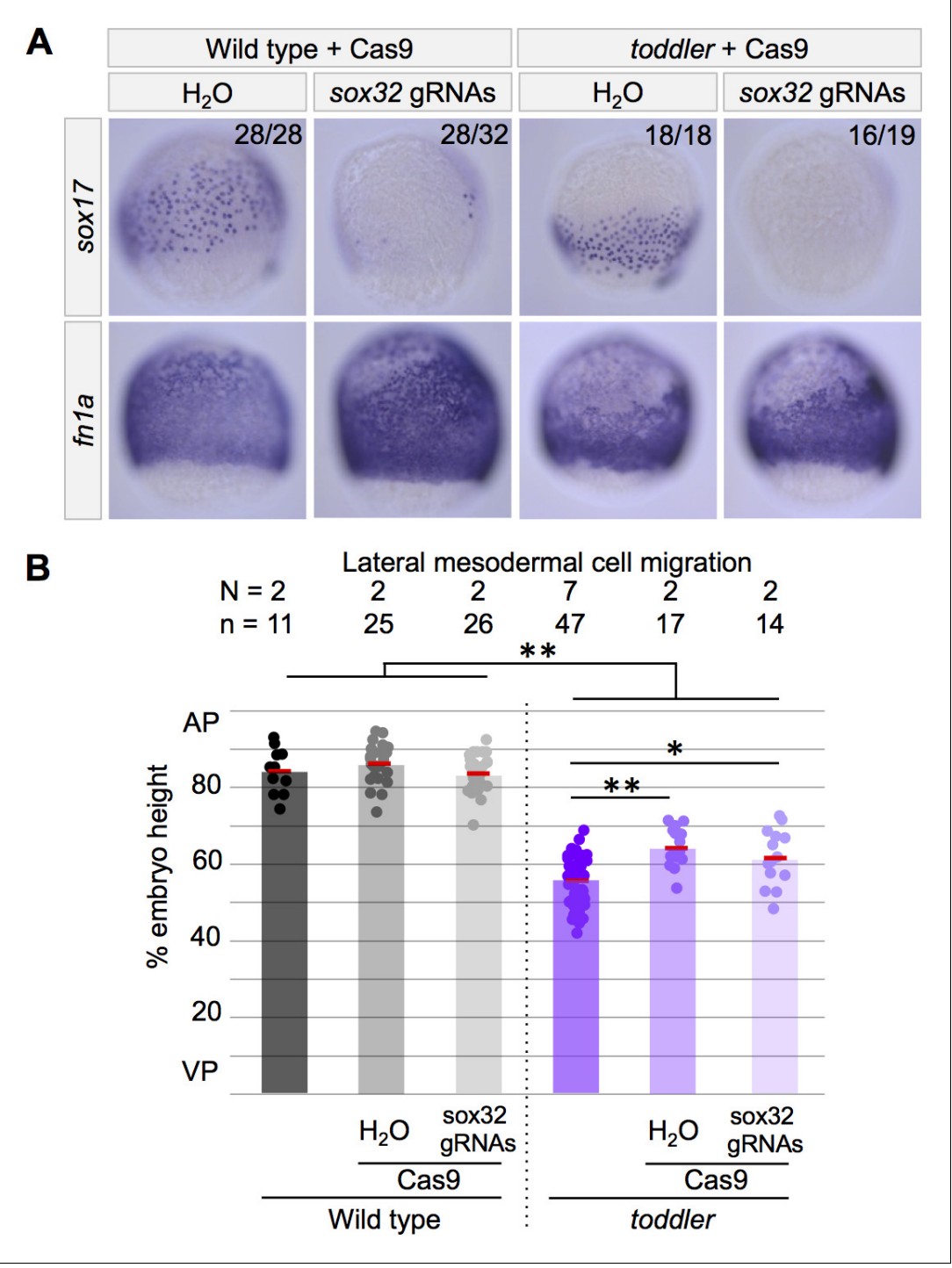

**Figure 6.** Toddler's role in mesodermal cell migration is independent of endoderm. The presence or absence of endoderm does not affect lateral mesodermal cell migration in wild-type and *toddler* mutants. Cas9-mediated mutagenesis was used to generate *sox32* mutants. Embryos were injected at the one-cell stage with Cas9 and *sox32* gRNAs. In situ hybridization for *sox17* and *fn1a* at 75% epiboly; dorsal to the right. (**A**) Representative images of embryos analyzed in B. (**B**) Quantification of mesodermal cell migration from embryos in A as described in *Figure 2A*. AP = Animal pole; VP = vegetal pole. Each point represents a single embryo. Red bars are averages; \*\*p<1.5×10$^{-6}$; \*p<0.05; unpaired two-tailed t-test.

DOI: https://doi.org/10.7554/eLife.22626.011

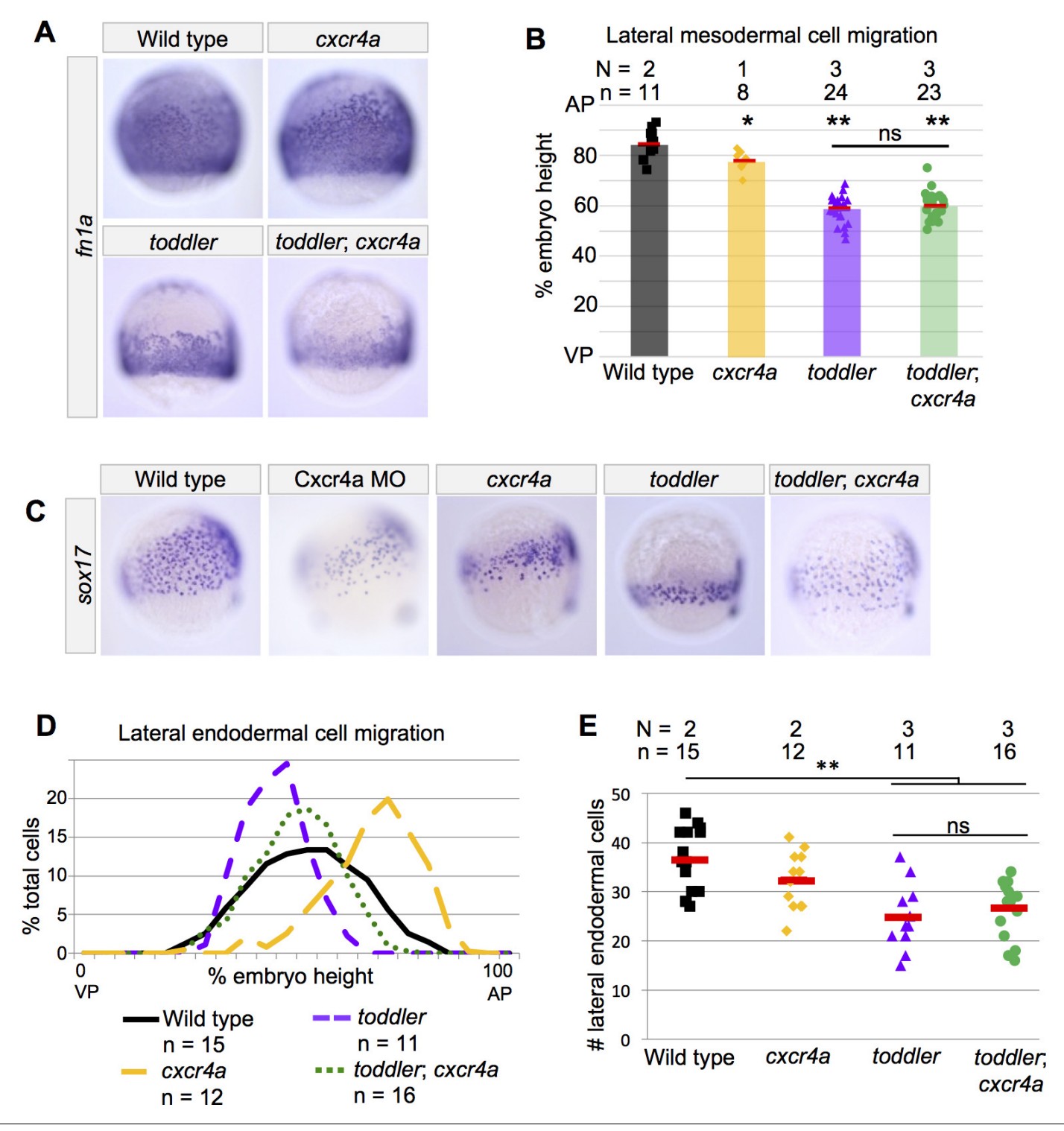

**Figure 7.** Endodermal cell migration does not require Toddler signaling in *cxcr4a* mutants. (A–B) Analysis of lateral mesodermal cell migration defects in *cxcr4a* mutant embryos by in situ hybridization for *fn1a*. Mesodermal cell migration is normal in stable *cxcr4a* single mutants. *toddler;cxcr4a* double mutants show mesodermal cell migration defects that resemble *toddler* single mutant siblings. (C) Representative images of embryos analyzed in D and E. In situ hybridization for *sox17* at 75% epiboly; dorsal to the right. *Cxcr4a* morphants and *cxcr4a* mutants have excessive animal pole-directed migration of vegetal endoderm. Endoderm patterning in *toddler;cxcr4a* double mutants resembles wild-type embryos more than *toddler* single mutant siblings. (D) Measurement of frequency with which cells were found at a given location in an embryo of a certain genotype. A cell at the animal pole corresponds to 100% embryo height, while a cell at the vegetal pole corresponds to 0%. AP = Animal pole=100%; VP = vegetal pole=0%. (E) The
*Figure 7 continued on next page*

*Figure 7 continued*

number of lateral endodermal cells is unchanged between *toddler* single and *toddler;cxcr4a* double mutants. Each point represents a single embryo. Red bars are averages. (**B and E**) ns: $p > 0.46$; *$p < 0.05$; **$p < 0.0005$; unpaired two-tailed t-test. N = number of independent experiments; n = number of embryos.

DOI: https://doi.org/10.7554/eLife.22626.012

analyzed as early as 60% epiboly. For endoderm, images were analyzed with the lateral side in focus and the dorsal side in the plane of the screen. ImageJ was used to identify the center of the lateral side 90° from dorsal and to measure the absolute height and width of the embryo. The relative locations of *sox17+* cells in relationship to embryo height were recorded for all cells falling on or within 10% of the embryo width from the center of the lateral side. The number of cells within this area was used as a proxy for endodermal cell number.

For mesoderm, images were analyzed with the dorsal side forward and cross sections of the lateral sides in focus. The relative locations of the most animal *fn1a+* cells in relationship to embryo height were recorded. All cell measurements were normalized to percent embryo height before comparison to other embryos. All *toddler;lefty2* or *toddler;cxcr4a* double mutants were compared to their own *toddler* single mutant siblings. Appropriate numbers of embryos to be analyzed were determined based on robustness of phenotypes across biological replicates and supported by measurements of statistical difference.

## RNAseq

Wild-type (TLAB) and *toddler* mutant zebrafish embryos were kept at 28C and staged according to standard procedures. Fifty embryos were collected per genotype per stage (sphere (4 hpf), early dome (5 hpf), germ ring (6 hpf), shield (7 hpf)). Total RNA was isolated using the standard TRIzol (Invitrogen, Waltham, Massachusetts) protocol. Genomic DNA was removed by TURBO-DNase treatment followed by phenol/chloroform extraction. The RNA quality was confirmed using the Agilent 2100 Bioanalyzer. RNAseq libraries were generated using the Illumina TruSeq RNA Library Prep Kit

**Table 1.** Primer and target sequences.

| Primers | | gRNA targets | |
|---|---|---|---|
| lefty1 1 | CGTGGCTTTCATGTATCACCTTC | cxcr4a target 1 | G G A CATCGGAGC CAACTTTG |
| lefty1 1 | G CATTAG CCTATATG TTAACTTG CAC. | cxcr4a target 2 | GTAC CG TCTG CA C CT CTCAG |
| lefty2 1 | GGGACACAAG CTTTG AAGG G | cxcr4a target 3 | CG C CTTC ATCA GTTTG G ACC |
| lefty2 2 | TCCCTGTGTGAGTGAGATCG | cxcr4a target 4 | CGCCGCGCTCCTCACTGTGC |
| lefty2 3 | CAGCTGTTCATTTTGACCACTCAC | cxcr4a target 5 | G GA CTC GTTTG TCACAT G G G |
| lefty2 4 | AT G GAG CTTC AG C AT G G AC AG | sox32 target 1 | CGTTCTGATGTTG C AAATAGTGG |
| aplnrB 1 | TGTGTGAATATGATGAGTGGGAAC | sox32 target 2 | G G CTTAAT G G GCC CGACGCGGGG |
| aplnrB 2 | AGTGGTATCCCAGAGCAGTGTAG | sox32 target 3 | CC GC G TCG G G CCC ATTAAG CC CG |
| cxcr4a 1 | CGTCTTTGAAGATGATTTATCAGC | sox32 target 4 | G TT C ATCAT G TG G AC G AAAG AG G |
| cxcr4a 2 | CA C GTAAATG ATG CG G TT G G | sox32 target 5 | CGAAGTGGTATGATGAAGAGTGG |
| cxcr4a 3 | AG ACT G AA G GAG CTG G AG AAG | sox32 target 6 | AGTG GAAAC GT GTTTG AT G GT G G |
| qPCR ta F | G AAC CAC AG AG GTG CTC CATATC | sox32 target 7 | G ACTCT GA GTAAGC AG ACC G TG G |
| qPCR ta R | CTGGTGTTGGAGGTAGTGTTTGTG | sox32 target 8 | GTAGAGCTCCATGATAGGTGGGG |
| qPCR leftyl F | AAG CT CTAC AAG AAG G CCC C AC ACAAG | sox32 target 9 | G CGTGTGTG CTGTGTTTG G GTG G |
| qPCR leftyl R | TTCG TG AATG G G AATC AAC CTG G A A | | |
| qPCR actin F | ATCAGGGTGTCATGGTTGGT | | |
| qPCR actin R | CACGCAGCTCGTTGTAGAAG | | |
| sox32 F | AAACTTCTCACG CTT C AC ACC | | |
| sox32 R | CCATCCAGATTGCTG CTG ATTT | | |

DOI: https://doi.org/10.7554/eLife.22626.013

v2. Libraries were sequenced on a HiSeq 2500 (single end 51 bp reads). Reads were aligned using TopHat v2.0.13 (*Trapnell et al., 2009*) with the following command for each sample 'tophat -o < output directory> -p 16 –no-novel-juncs -G < gene table><Bowtie2 genome index><fastq reads>". Transcript abundance and differential expression were determined using Cufflinks v2.2.1 (RRID:SCR_014597)(*Trapnell et al., 2012*) with the following command for each developmental stage "cuffdiff -p 16 -b < genome.fa -u -L < labels > o<output directory><gene table><wild type aligned reads. bam file><toddler mutant aligned reads. bam file>". Plots of differential gene expression were generated in Rstudio. The RNA-Seq data set was deposited to GEO; GEO acquisition number GSE89319.

## Transplants

Transplantations were carried out using a syringe and a micromanipulator. Donor embryos were injected at the one-cell stage with 50 pg *gfp* mRNA and a given amount of *ndr1* or *ndr2* mRNA. 10–20 cells were transplanted at sphere stage from the animal pole of the donor to the animal pole of the uninjected host. All embryos were stage matched and transplantations were homotypic: wild type into wild type or *toddler* mutant into *toddler* mutant. Embryos were collected 1.5 hr later at 50% epiboly. Following in situ hybridization for *ta* or *gsc*, GFP protein in transplanted cells was probed for using anti-GFP-POD and the VWR DAB reagent set (VWR Catalog number 95059–296, Radnor, Pennsylvania). Embryos were imaged in glycerol. The diameter of induction was measured using ImageJ. When potential differences in phenotype were present experiments were repeated to confirm presence of the differential phenotype across biological replicates.

## qRT-PCR

Wild-type and *toddler* mutants were injected at the one-cell stage with water or increasing amounts of *ndr1* mRNA. Embryos were stage matched at the 16 cell stage, collected 4.25 hr later and frozen in liquid nitrogen. Total RNA from 8 to 18 embryos was extracted using Omega E.Z.N.A Total RNA Kit (Norcross, Georgia). cDNA was synthesized using BioRad iScript kit (Hercules, California). qPCR was carried out for technical and biological replicates using GoTaq on a CFX96 machine. $2^{-\Delta\Delta Ct}$ was calculated for each target gene using actin as a reference. Biological replicates were obtained from separate experimental set-ups. Technical replicates were obtained by splitting biological replicates into two samples after the cDNA prep but before carrying out qPCR. Necessary sample size was determined based on robustness of phenotype and consistency across biological replicates.

## Lightsheet microscopy

Gastrulation movements were analyzed by lightsheet microscopy in *drl:eGFP* (*Video 1*) and *sox17: eGFP* transgenic wild-type and *toddler* mutant embryos. Imaging was performed as outlined in *Pauli et al., 2014*. In brief, embryos were allowed to develop at 28°C until early dome stage. Within a single experiment, 2–5 embryos (containing at least one embryo of each genotype) were mounted in 1% low-melting point agarose in 1x Danieau's in glass capillaries (Zeiss, Oberkochen, Germany). Image acquisition was performed at a Lightsheet Z.1 microscope (Zeiss) at 26.5°C in a fish-water-filled imaging chamber. Time-lapse movies of up to five embryos were recorded in the multi-view mode at time-intervals between frames no larger than 180 s.

Timelapse acquisition settings for imaging on the Lightsheet Z.1 (Zeiss) (Single Plane Illumination System): 20x/1.0 water immersion objective, 0.5x zoom; 488 nm laser; Detection: Dual PCO.Edge second-generation sCMOS cameras, liquid cooled, 1920 × 1920 pixels,~70 fps; 16 bit images (1920 × 1920 pixels; 878.8 μm x 878.8 μm; pixel size 0.46 μm); Dual-side illumination with online dual fusion and pivot scan; z-stacks: 80–190 slices per embryo; 1.6–2.5 μm intervals.

## Endodermal and mesodermal cell tracking

Manual tracking of *sox17:eGFP* and *drl:eGFP*-positive cells was done in FIJI by hand (endoderm) or using TrackMate (mesoderm). For endodermal cells, a window of 20–65 non-dorsal *sox17:eGFP* + cells were monitored for cell divisions from the onset of GFP expression to approximately 85% epiboly in three time windows to allow for correction of focus (20 frames, 54 frames, 20 frames, respectively). Intervals between frames was no more than 3.5 min. Percent proliferation was determined by dividing the number of splitting events by the number of cells present at the beginning of

the given time window. Proliferation events were averaged over all three time windows for each embryo.

For mesodermal cells, embryos were aligned at the beginning of internalization and individual GFP-positive cells were manually tracked until approximately 85% epiboly. Intervals between frames was no more than 3 min. TrackMate data was exported to R for computational analysis. Data from each embryo were binned into four time windows of equal length based on real time. For each measurement, only cells with 5 of more data points per time window were utilized and the entire first time window was excluded from further analysis. Straightness was determined by dividing the total displacement by the total distance. Total displacement was determined by summing the displacement made by a cell during each frame in a time window. Total distance was measured as the distance between the cell's location in the first and last frame of a time window. Velocity was measured by dividing the total displacement of a cell by the time for each time window. Rose plots were generated by plotting the angle moved weighted by the displacement traveled by each cell for each frame across a time window.

## TUNEL staining

Dechorionated embryos were collected at various time points in 4% paraformaldehyde, fixed at 4° overnight, then dehydrated into methanol (similar to collection for in situ hybridization) and stained using the In Situ Cell Death Detection Kit, TMR red, from Sigma Aldrich (St Louis, Missouri). Embryos were mounted in low-melting point agarose and one lateral side of each embryo was imaged using an LSM 880 confocal microscope with a < 1 micron pixel width and <4.5 micron voxel-depth. Maximum intensity projections were analyzed and TUNEL+ cells were defined as a single, large positive spot or multiple, smaller spots in an aggregate.

## Acknowledgements

The authors thank Wiebke Herzog for her generous gift of *aplnrA* mutants. The authors also thank Jeff Farrell and Adam Norris for critical reading of the manuscript and Kathryn Berg for her assistance with TUNEL staining.

## Additional information

### Funding

| Funder | Grant reference number | Author |
|---|---|---|
| National Institutes of Health | F31HD081925 | Megan L Norris |
| National Institutes of Health | HD076935 | Andrea Pauli |
| Boehringer Ingelheim | | Andrea Pauli |
| Human Frontier Science Program | CDA00066/2015 | Andrea Pauli |
| American Cancer Society | | James A Gagnon |
| National Institutes of Health | R01 GM056211 | Alexander F Schier |
| Arnold and Mabel Beckman Foundation | | Nathan D Lord |

The funders had no role in study design, data collection and interpretation, or the decision to submit the work for publication.

### Author contributions

Megan L Norris, Conceptualization, Resources, Data curation, Formal analysis, Investigation, Visualization, Methodology, Writing—original draft, Writing—review and editing; Andrea Pauli, Conceptualization, Resources, Data curation, Methodology, Writing—review and editing; James A Gagnon, Resources, Data curation, Formal analysis, Methodology, Writing—review and editing; Nathan D Lord, Formal analysis, Visualization, Methodology, Writing—review and editing; Katherine W Rogers,

Conceptualization, Resources, Formal analysis; Christian Mosimann, Leonard I Zon, Resources, Contributed essential reagents; Alexander F Schier, Conceptualization, Supervision, Funding acquisition, Writing—review and editing

### Author ORCIDs
Megan L Norris  http://orcid.org/0000-0003-0666-2665
Andrea Pauli  http://orcid.org/0000-0001-9646-2303
Christian Mosimann  http://orcid.org/0000-0002-0749-2576
Leonard I Zon  https://orcid.org/0000-0003-0860-926X
Alexander F Schier  http://orcid.org/0000-0001-7645-5325

### Ethics
Animal experimentation: This study adhered to recommendations put forth in the Guide for the Care and Use of Laboratory Animals of the National Institutes of Health. Protocol #25-08 of Harvard University was approved by the Institutional Animal Care and Use Committee (IACUC).

### Decision letter and Author response
Decision letter https://doi.org/10.7554/eLife.22626.017
Author response https://doi.org/10.7554/eLife.22626.018

## Additional files

### Supplementary files
• Transparent reporting form
DOI: https://doi.org/10.7554/eLife.22626.014

### Major datasets
The following dataset was generated:

| Author(s) | Year | Dataset title | Dataset URL | Database, license, and accessibility information |
|---|---|---|---|---|
| Gagnon JA,  Pauli A,  Schier AF | 2016 | RNAseq of wild type and Toddler mutant zebrafish embryos | https://www.ncbi.nlm.nih.gov/geo/query/acc.cgi?acc=GSE89319 | Publicly available after publication at NCBI Geo (Accession no: GSE89319) |

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
