## [Decision Letter]

[Editors’ note: a previous version of this study was rejected after peer review, but the authors submitted for reconsideration. The first decision letter after peer review is shown below.]

Thank you for submitting your work entitled "Toddler signaling regulates mesodermal cell migration downstream of Nodal signaling" for consideration by *eLife*. Your article has been reviewed by three peer reviewers, one of whom is a member of our Board of Reviewing Editor, and the evaluation has been overseen by a Senior Editor. The reviewers have opted to remain anonymous.

Our decision has been reached after consultation between the reviewers. Based on these discussions and the individual reviews below, we regret to inform you that your work will not be considered further for publication in *eLife*.

The reviewers all agreed that the function of Toddler in regulating Nodal signalling was an interesting topic. They also acknowledged that it has been controversial and that this manuscript aims to address the two proposed models in a precise and targeted manner. However, the results were found to be currently too preliminary to allow a firm conclusion to be drawn about the precise role of Toddler.

Reviewer #1:

This paper addresses an ongoing dispute about the function of the Toddler peptide in controlling mesendodermal development during zebrafish gastrulation. Specifically the disputed question is whether Toddler acts early, to enhance Nodal signaling, and therefore to specify endoderm cells; or, alternatively, whether it acts later, to control the migration of mesendoderm.

This elegant paper addresses three predictions that may distinguish the two models. All three sets of experiments lead the authors to conclude that Toddler acts downstream of Nodal signaling to control the migration mesodermal cells and, secondarily, the migration of normally attached endodermal cells. The data are clear, and the conclusions, while quite narrow, appear well supported. That may be slightly different from saying that this paper absolutely settles the case: it is possible that more complex hybrid models would be consistent with all results.

The authors discuss the discrepancies with the earlier conclusion that favors the endodermal specification model, and suggest reasons that could account for the distinct conclusions.

Reviewer #2:

The manuscript by Norris et al. investigates the mechanism by which the peptide Elabela/Toddler/Apela regulates mesendoderm development. Unlike the zebrafish cardia bifida mutants (cas, mil, fau) that also have reduced mesendoderm and endoderm cell migration defects, or nodal signaling mutants (cyc;sqt or oep) that have mesendoderm specification defects, in toddler mutants Nodal signaling is intact and mesendoderm specification is initially normal. Yet, there is a reduction in endoderm cells by mid-gastrula stages, mesoderm and endoderm cells do not migrate correctly, and toddler mutants have very small or no hearts. Using loss of lefty inhibitor to increase nodal signalling (and endoderm specification) in the context of toddler mutants, the authors tested and rule out the "specification model" which proposed a role for toddler as an enhancer of Nodal signaling. This part of the manuscript is, for the most part, nicely done.

Norris and coworkers then tested the "migration model" by loss of the GPCR/chemokine receptor, cxcr4a and suggest that Toddler indirectly affects endoderm migration via Cxcr4 signaling. This part of the manuscript is not very convincing: the tether postulate can only partially explain the cell migration defect (in the endoderm), and it is unclear how mesoderm migration is affected in toddler mutants. In addition, why are endoderm cell numbers reduced? Do these cells die or is there delayed/other cell division defects in the mutant mesendoderm?

Are the authors certain that initial mesendoderm specification is normal in toddler mutants? Norris and coworkers show that response to Nodal signalling is intact in toddler embryos, however both the Reversade and Schier groups have previously shown that in toddler mutants, the scl and gata5 expression domains are expanded. How can this be explained?

If the primary function for toddler is in regulation of cell migration, it might be useful to investigate individual cell behaviours, rather than exclusively rely upon the expression of germ layer markers. For instance, is protrusive activity altered in mutant cells (which might then affect cell migration)? Are there defects in cytoskeletal organization in toddler mutant mesendoderm, which might affect cell division and cell numbers as well as cell migration?

The authors examined RNAseq data and conclude that the gene expression profile is largely unchanged in toddler mutant embryos compared to controls, at least at early embryonic stages (sphere-shield). However, for abundant molecules (such as those comprising the cytoskeleton) and for those that have a significant maternal contribution, the subcellular distribution of the proteins might be important, but may not be evident from absolute transcript levels.

In summary, the manuscript has some novel findings; the data ruling out effects of Toddler on mesendoderm specification via Nodal signaling is interesting and contrasts with a recent report by the Reversade and Scott groups. However, the manuscript does not satisfactorily address how Toddler affects cell migration (especially the mesoderm). In its present form, the manuscript seems more appropriate for a specialized audience.

Reviewer #3:

This paper seeks to distinguish between two models for Toddler function in zebrafish embryos. One model suggests that Toddler is required for Nodal signalling and thus is required to correctly specify endoderm. The other model postulates that Toddler regulates mesendodermal cell migration downstream of Nodal signaling. The authors' data do not support the first model, but give some support to the second model. However, in general, I think that the work is preliminary, and key controls are missing in my view. There is also no insight into how Toddler could be functioning to promote mesodermal cell migration.

1) In Figure 1 and Figure 2, the authors compare the toddler mutant and the lefty2;toddler double mutant, without showing data for the lefty2 mutant alone. This must be shown in order to be able to assess the effect of mutating toddler in the lefty2 mutant background.

2) In Figure 2, the heart sizes should be quantitated. It is not sufficient to present the percentage of embryos with smaller hearts.

3) In general I think the images showing sox17 staining to quantitate numbers of endodermal cells are low in quality. It would be better to perform fluorescent in situs, which would be easier to image and quantitate.

4) The loss of function experiments for sox32 have been performed using transient expression of guide RNAs and Cas9. There are no controls showing that sox32 has been deleted. The authors must prove what deletions they are achieving in these embryos and they must prove the loss of sox32. It would be much better if they used a stable line as they have for the cxcr4a mutant.

5) There is no evidence presented that the lefty2 mutants or the cxcr4a mutants are nulls.

6) The authors say that loss of sox32 or overexpression of sox32 has no effect on aplnrB expression. The data in Figure 5 do not agree with this. There is certainly lower expression when sox32 is overexpressed, and higher expression when sox32 is deleted. In addition, what is the anterior staining in Figure 5 in the sox32 gRNAs+cas9? In Figure 5—figure supplement 1, sox32 deletion also seems to increase fn1a staining.

6) In Figure 6, it really is not clear that in the cxcr4a null the sox17 cells migrate further animally. This needs to be investigated further.

[Editors’ note: what now follows is the decision letter after the authors submitted for further consideration.]

Thank you for resubmitting your work entitled "Toddler signaling regulates mesodermal cell migration downstream of Nodal signaling" for further consideration at *eLife*. Your revised article has been favorably evaluated by Marianne Bronner (Senior editor), a Reviewing editor, and three reviewers.

The manuscript has been improved but there are some remaining issues that need to be addressed before acceptance, as outlined below:

1) It is essential to determine the fate of the missing endodermal cells: is there a change in proliferation or cell death? The authors should use the tools they described in Pauli et al. 2014 to address this key question.

2) It would be helpful to readers to include a more measured and constructive discussion about the differences between the present results and those described by Deshwar et al. Different reagents and approaches were used and it is not clear whether these could account for at least some of the differences.

3) With regard to quantifying the heart size in Figure 2, the authors argue that it is difficult to do this given the in situ images and the irregular shape of the heart. This may be the case, but it is essential to define what constitutes a small heart. Thus, some level of quantification is required.

4) It is important to demonstrate that the lefty mutants they use do actually lead to elevated Nodal signaling

---

## [Author Response]

[Editors’ note: the author responses to the first round of peer review follow.]

The reviewers all agreed that the function of Toddler in regulating Nodal signalling was an interesting topic. They also acknowledged that it has been controversial and that this manuscript aims to address the two proposed models in a precise and targeted manner. However, the results were found to be currently too preliminary to allow a firm conclusion to be drawn about the precise role of Toddler.

In response to the helpful suggestions of the reviewers, we have provided additional results concerning the role of Toddler in regulating mesodermal cell migration. From our previous data it was not clear whether directionality, speed, persistence of migration, or aspects of each cause the migration defects observed in *toddler* mutants. Through live-cell imaging and analysis of mesodermal cells during gastrulation in wild-type and *toddler* mutant embryos we are now able to conclude that (Abstract): “Mesodermal cell migration defects in *toddler* mutants result from a decrease in animal pole-directed migration and are independent of endoderm. Conversely, endodermal cell migration defects are dependent on a Cxcr4a-regulated tether to mesoderm”.

We address the other questions raised by the reviewers in detail below.

Reviewer #1:This paper addresses an ongoing dispute about the function of the Toddler peptide in controlling mesendodermal development during zebrafish gastrulation. Specifically the disputed question is whether Toddler acts early, to enhance Nodal signaling, and therefore to specify endoderm cells; or, alternatively, whether it acts later, to control the migration of mesendoderm.This elegant paper addresses three predictions that may distinguish the two models. All three sets of experiments lead the authors to conclude that Toddler acts downstream of Nodal signaling to control the migration mesodermal cells and, secondarily, the migration of normally attached endodermal cells. The data are clear, and the conclusions, while quite narrow, appear well supported. That may be slightly different from saying that this paper absolutely settles the case: it is possible that more complex hybrid models would be consistent with all results.The authors discuss the discrepancies with the earlier conclusion that favors the endodermal specification model, and suggest reasons that could account for the distinct conclusions.

We thank the reviewer for her/his strong support.

Reviewer #2:The manuscript by Norris et al. investigates the mechanism by which the peptide Elabela/Toddler/Apela regulates mesendoderm development. Unlike the zebrafish cardia bifida mutants (cas, mil, fau) that also have reduced mesendoderm and endoderm cell migration defects, or nodal signaling mutants (cyc;sqt or oep) that have mesendoderm specification defects, in toddler mutants Nodal signaling is intact and mesendoderm specification is initially normal. Yet, there is a reduction in endoderm cells by mid-gastrula stages, mesoderm and endoderm cells do not migrate correctly, and toddler mutants have very small or no hearts. Using loss of lefty inhibitor to increase nodal signalling (and endoderm specification) in the context of toddler mutants, the authors tested and rule out the "specification model" which proposed a role for toddler as an enhancer of Nodal signaling. This part of the manuscript is, for the most part, nicely done.

We thank the reviewer for her/his strong support of the part of our study that focuses on Nodal signaling.

Norris and coworkers then tested the "migration model" by loss of the GPCR/chemokine receptor, cxcr4a and suggest that Toddler indirectly affects endoderm migration via Cxcr4 signaling. This part of the manuscript is not very convincing: the tether postulate can only partially explain the cell migration defect (in the endoderm), and it is unclear how mesoderm migration is affected in toddler mutants.

We agree that it was unclear what aspect of migration, such as directionality, speed, or persistence of migration, caused migration defects. To address this question, we performed live-cell imaging of mesodermal cells during gastrulation in wild-type and *toddler* mutant embryos (new Figure 5, Video 1). Our results reveal that mesodermal cells in *toddler* mutants migrate more slowly than wild type and have diminished animal-pole directed migration during gastrulation.

In addition, why are endoderm cell numbers reduced? Do these cells die or is there delayed/other cell division defects in the mutant mesendoderm?

The decreased number of endodermal cells is indeed interesting and perplexing. We have included a paragraph in our discussion to address this remaining question: “The interplay between mesoderm and endoderm might modulate endodermal cell number, as has been suggested in mouse embryonic stem cells (Cheng et al. 2013). This fine-tuning may be disrupted when mesoderm fails to migrate properly. It is also conceivable that Toddler signaling promotes endoderm proliferation during gastrulation. In either case, our results show that increasing endodermal cell number does not rescue migration or other defects in *toddler* mutants”.

Are the authors certain that initial mesendoderm specification is normal in toddler mutants? Norris and coworkers show that response to Nodal signalling is intact in toddler embryos, however both the Reversade and Schier groups have previously shown that in toddler mutants, the scl and gata5 expression domains are expanded. How can this be explained ?

The reviewer is correct that Nodal signaling is paramount to proper mesendoderm specification. For this reason we included RNA-sequencing and *in situ* hybridization data that show normal mesendoderm specification in *toddler* mutants (Figure 3). We have also previously shown that endoderm (*sox17*), mesoderm (*fn1, ta*) and ectoderm (*gsc*) appear normal during early stages of gastrulation in *toddler* mutants (Pauli et al. 2014 Suppl Figure S10). Comparatively, *scl* is ectopically expressed at much later stages, possibly as a secondary defect caused by abnormal gastrulation mutants (Pauli et al. 2014 Figure 2). We are not aware of data that shows that *gata5* expression domains are expanded.

If the primary function for toddler is in regulation of cell migration, it might be useful to investigate individual cell behaviours, rather than exclusively rely upon the expression of germ layer markers. For instance, is protrusive activity altered in mutant cells (which might then affect cell migration)? Are there defects in cytoskeletal organization in toddler mutant mesendoderm, which might affect cell division and cell numbers as well as cell migration?The authors examined RNAseq data and conclude that the gene expression profile is largely unchanged in toddler mutant embryos compared to controls, at least at early embryonic stages (sphere-shield). However, for abundant molecules (such as those comprising the cytoskeleton) and for those that have a significant maternal contribution, the subcellular distribution of the proteins might be important, but may not be evident from absolute transcript levels.

We agree with the reviewer that the precise mechanism by which Toddler regulates migration is an intriguing question. We have added live-cell imaging of mesodermal cells during gastrulation (Figure 5, Video 1), which shows that Toddler is required for proper migration towards the animal pole as well as proper speed of migration. We do not yet understand the molecular mechanism regulating the directionality and speed changes observed in *toddler* mutants. A detailed analysis of cytoskeletal changes is difficult for technical reasons. First, there is currently a lack of genetic tools for specifically manipulating mesoderm during gastrulation. Second, there have been many attempts in the past decade to find the downstream target of the Apelin receptors, but each method failed to conclusively identify a well defined pathway (for example Deng et al. 2015; He, Chen, and Li 2016; Paskaradevan and Scott 2012) We hope to address this question in future studies that are beyond the scope of this paper.

In summary, the manuscript has some novel findings; the data ruling out effects of Toddler on mesendoderm specification via Nodal signaling is interesting and contrasts with a recent report by the Reversade and Scott groups. However, the manuscript does not satisfactorily address how Toddler affects cell migration (especially the mesoderm). In its present form, the manuscript seems more appropriate for a specialized audience.

We thank the reviewer for her/his support and refer her/him to the end of this response, where we discuss in more detail why we believe that *eLife* is the best forum for this study. In addition, the new mesoderm migration data should make the paper more conclusive.

Reviewer #3:This paper seeks to distinguish between two models for Toddler function in zebrafish embryos. One model suggests that Toddler is required for Nodal signalling and thus is required to correctly specify endoderm. The other model postulates that Toddler regulates mesendodermal cell migration downstream of Nodal signaling. The authors' data do not support the first model, but give some support to the second model. However, in general, I think that the work is preliminary, and key controls are missing in my view. There is also no insight into how Toddler could be functioning to promote mesodermal cell migration.1) In Figure 1 and Figure 2, the authors compare the toddler mutant and the lefty2;toddler double mutant, without showing data for the lefty2 mutant alone. This must be shown in order to be able to assess the effect of mutating toddler in the lefty2 mutant background.

We thank the reviewer for pointing out this inadvertent omission. Data for the *lefty2* single mutant has now been added to Figure 1 and 2.

2) In Figure 2, the heart sizes should be quantitated. It is not sufficient to present the percentage of embryos with smaller hearts.

This figure was included to compare our results to the Deshwar et al. *eLife* paper, which used the same analysis. Exact quantification of heart sizes by absolute numbers is difficult based on in situ images and the irregular shape of the heart. But given the clear differences between *toddler* mutants and wild-type embryos, a more detailed measurement is unlikely to affect the conclusions drawn from this experiment.

3) In general I think the images showing sox17 staining to quantitate numbers of endodermal cells are low in quality. It would be better to perform fluorescent in situs, which would be easier to image and quantitate.

The reviewer’s concern was likely due to poor resolution of the PDF-converted small figure.

4) The loss of function experiments for sox32 have been performed using transient expression of guide RNAs and Cas9. There are no controls showing that sox32 has been deleted. The authors must prove what deletions they are achieving in these embryos and they must prove the loss of sox32. It would be much better if they used a stable line as they have for the cxcr4a mutant.

We thank the reviewer for asking us to provide additional support for our sox32 loss-of-function experiment. We have included data in Figure 5—figure supplement 1 that shows the high efficacy of injected Cas9 and *sox32* gRNAs in mutating the *sox32* locus. The phenotype of these Cas9+*sox32* gRNA injected embryos matches the known *casanova (sox32*) mutant phenotype in the complete loss of endoderm (Figure 6)(Alexander et al. 1999).

5) There is no evidence presented that the lefty2 mutants or the cxcr4a mutants are nulls.

The *lefty2* mutation is a null allele: an 11-nucleotide deletion that results in an out of frame inactive coding sequence. This result will be published in Rogers et al. (in preparation) but we would be happy to share this data with the reviewer. We have also added more information about the *cxcr4a* mutant. This allele deletes 433 bp of a 1083 bp long coding sequence, rendering the GPCR non-functional. Moreover, the gastrulation phenotype of the homozygous *cxcr4a* mutant recapitulates the morphant phenotype (Figure 7) (Mizoguchi *et al.*, 2008; Nair and Schilling, 2008).

6) The authors say that loss of sox32 or overexpression of sox32 has no effect on aplnrB expression. The data in Figure 5 do not agree with this. There is certainly lower expression when sox32 is overexpressed, and higher expression when sox32 is deleted. In addition, what is the anterior staining in Figure 5 in the sox32 gRNAs+cas9? In Figure 5—figure supplement 1, sox32 deletion also seems to increase fn1a staining.

We apologize for the lack of clarity. We have revised subsection “Toddler receptors are expressed in mesodermal cells” and the legends Figure 5 and Figure 5—figure supplement 1 to say that *aplnrB*, *aplnrA, fn1a* and *ta* decrease slightly or do not change in the presence of excess endoderm and increase in the absence of endoderm.

6) In Figure 6, it really is not clear that in the cxcr4a null the sox17 cells migrate further animally. This needs to be investigated further.

We apologize for the ambiguity in our word choice. We have changed our wording to clarify that the cells that normally reside more vegetally in wild type embryos migrate more animally in the mutants, consistent with previous Cxcr4a morphant studies (Mizoguchi *et al.*, 2008; Nair and Schilling, 2008).

[Editors' note: the author responses to the re-review follow.]

The manuscript has been improved but there are some remaining issues that need to be addressed before acceptance, as outlined below:1) It is essential to determine the fate of the missing endodermal cells: is there a change in proliferation or cell death? The authors should use the tools they described in Pauli et al. 2014 to address this key question.

We have more closely examined the fate of the missing endodermal cells in *toddler* mutant embryos (Figure 1—figure supplement 1). We first confirmed that similar numbers of endodermal cells are present at 60% epiboly in wild-type and *toddler* mutant embryos and that the number of cells increases only in wild type (Figure 1—figure supplement 1). To investigate a potential proliferation defect in *toddler* mutant embryos, we tracked division rates of *sox17*:GFP positive endodermal cells during gastrulation from light sheet movies. We found that wild type and *toddler* mutant endodermal cells proliferate at the same rate during gastrulation (Figure 1—figure supplement 1). However, while tracking proliferation, we observed endodermal cells dying in *toddler* mutants (Figure 1—figure supplement 1). To determine whether cell death was elevated in *toddler* mutant embryos, we used TUNEL staining and found that the number of TUNEL+ cells was increased throughout gastrulation in *toddler* mutants compared to wild-type embryos (Figure 1—figure supplement 1 C,D). Together, this new data suggests that the missing endodermal cells observed at the end of gastrulation (Figure 1) are not due to initial specification or proliferation defects but instead may result from an increase in cell death in *toddler* mutants during gastrulation. This is consistent with other work, which observed Toddler had an anti-apoptotic role in human embryonic stem cells (Ho et al. *Cell Stem Cell,* 2015). We have also updated the text by adding a paragraph that discusses the fate of the missing endodermal cells. We would like to emphasize that, though the decreased endodermal cell number is an interesting phenomenon, it is not the primary role of Toddler signaling. This is demonstrated in Figure 1 and Figure 2, whereby restoring endodermal cell numbers does not abrogate the *toddler* mutant phenotype.

2) It would be helpful to readers to include a more measured and constructive discussion about the differences between the present results and those described by Deshwar et al. Different reagents and approaches were used and it is not clear whether these could account for at least some of the differences.

The reviewers are correct in noting differences in the experimental approaches between our paper and the Deshwar et al. paper. We have changed our discussion to more explicitly discuss how these differences may have affected our results and make direct comparison between the two studies difficult. Of the three suggested possibilities, we favor the hypothesis that the morpholinos cause off target effects and toxicity for the following reasons. 1.) There is currently no evidence for differential roles of Toddler and Apelin receptor signaling during zebrafish development. 2.) Unlike other disagreeing morpholino/mutant pairs, in which the mutant lacks all or most of the morphant phenotype, *apelin receptor a and b* mutants and morphants have similar defects in endoderm and mesoderm development during gastrulation and in heart development (Deshwar et al. 2016). We now write: “Our findings about Toddler signaling differ from a recent study that concluded Apelin receptor signaling enhances Nodal signaling (Deshwar et al. 2016). Whereas Deshwar and colleagues reported that apelin receptor a and b double morphants appear to have a delayed onset of Nodal signaling, we find that toddler mutants as well as apelin receptor a and b single and double mutants establish Nodal signaling normally (Figure 3). There are multiple possibilities that may explain these different results. First, it is possible that Toddler and/or the Apelin receptors have independent functions (Ho et al. 2015). Second, although apelin receptor mutants and morphants have similar morphological defects (Deshwar et al. 2016), knockdown of the receptors via morpholinos might reveal a phenotype that is masked or compensated in the null mutants (Rossi et al. 2015; Wei et al. 2016). Third, morpholino injection may cause non-specific delayed development or toxicity, as suggested by the epiboly defects seen in apelin receptor a morphants but not apelin receptor a mutants (Deshwar et al. 2016; Kok et al. 2015; Ekker & Larson 2001).”

3) With regard to quantifying the heart size in Figure 2, the authors argue that it is difficult to do this given the in situ images and the irregular shape of the heart. This may be the case, but it is essential to define what constitutes a small heart. Thus, some level of quantification is required.

We agree that it is important to clearly define what constitutes a small or normal sized heart. We have expanded our description of how we quantify heart size in Figure 2 and included more example images. We now write: “Hearts were classified as small if shortened by more than half of normal length and/or excessively narrow or thin. Hearts that were neither thin nor short but appeared to have looping defects or other patterning defects were classified as normal. In the case of toddler and toddler;lefty2 mutants, hearts were scored blind to genotype.”

4) It is important to demonstrate that the lefty mutants they use do actually lead to elevated Nodal signaling

We understand that the use of novel mutants warrants more careful analysis of phenotype and thank the reviewers for the opportunity to better characterize them. The *lefty* mutants are more completely characterized in a separate paper that we just resubmitted to *eLife* (Rogers et al.). This paper shows that the allele we study is a null allele and that both Lefty1 and Lefty2 have overlapping roles as inhibitors of Nodal signaling. Antibody staining also shows increased phosphorylation of the Nodal signal transducer, Smad2, in *lefty2* mutant embryos as compared to wild-type embryos (Author response image 1)(Dubrulle et al. 2015; Schier 2009). These results, coupled with the increase in Nodal target gene expression (*sox17,*
Figure 1), demonstrate that loss of *lefty2* leads to increased Nodal signaling.

**Author response image 1. respfig1:** Nodal signaling is elevated in *lefty*2 mutants. Smad2 is the intracellular effector of Nodal signaling and is phosphorylated when activated (Schier 2009). We measured phosphorylated Smad2 (pSmad2) by immunostaining as described in a forthcoming manuscript (Rogers et al. submitted). Representative cross-sections from pools of 3-6 embryos at 50% epiboly stage are shown. Left column: nuclei are labeled with Sytox green DNA stain. Middle column: α -pSmad2 immunostaining. Right column: magnified view of embryo margins highlighted in middle column with white boxes.